# Protective effects of clinical anticholinergic and anticholinesterase agents against *Bungarus multicinctus* venom and neurotoxin-rich snake venoms

Guowen Zhang[1,2◉], Mengqi He[2◉], Tongyi Sun[1]*, Wen-Hui Lee[2]*

**1** College of Life Science and Technology, Shandong Second Medical University, Weifang, Shandong, China, **2** Kunming Institute of Zoology, Chinese Academy of Sciences, Kunming, Yunnan, China

◉ These authors contributed equally to this work.
* tysun@sdsmu.edu.cn (TS); leewh@mail.kiz.ac.cn (W-HL)

## Abstract

*Bungarus multicinctus* is one of the most venomous and lethal snake species in mainland China, with envenomation resulting in a mortality rate as high as 23%. Currently, antivenin against *B. multicinctus* remains the gold standard for treating bites from this species. However, in remote mountainous areas or rural regions of China, access to antivenin may be delayed or unavailable following a bite. The lethal components of *B. multicinctus* venom include α-, β-, γ- and κ-bungarotoxins, which act on either the presynaptic or postsynaptic membrane, thereby altering acetylcholine levels in the synaptic cleft. This study aims to evaluate pharmaceutical agents that confer prophylactic or protective effects against bungarotoxins and neurotoxin-rich snake venoms. Based on theoretical considerations, we postulate that certain anticholinergic or anticholinesterase agents may exhibit potential protective efficacy against these venom components. This study investigated the protective effects of clinically standard anticholinergic and anticholinesterase agents—scopolamine, neostigmine methylsulfate, bethanechol chloride, pilocarpine hydrochloride, and atropine sulfate—against different bungarotoxins and wide distribution neurotoxin-rich crude snake venoms (*B. multicinctus*, *Naja atra* and *Ophiophagus hannah*). Our results showed that neostigmine methylsulfate and atropine sulfate exerted significant protective effects (P<0.01) against three-finger toxins contained in *B. multicinctus* venom, including α-bungarotoxin, γ-bungarotoxin, α-bungarotoxin+γ-bungarotoxin combinations, β-bungarotoxin+α-bungarotoxin combinations, and β-bungarotoxin+γ-bungarotoxin combinations. In mice animal models treated with *B. multicinctus* antivenin, neostigmine methylsulfate and atropine sulfate retained profound protective effects (P<0.01) against β-bungarotoxin+α-bungarotoxin and β-bungarotoxin+γ-bungarotoxin mixtures. However, these clinic-used drugs showed no significant protective effect against *B. multicinctus* crude venom, with only modest survival time

**Data availability statement:** All relevant data are in the manuscript and its supporting information files.

**Funding:** This study was supported by the National Key Research and Development Program of China (Grant No. 2023YFF1304900, awarded to WH-L) and the Project of Yunnan Provincial Department of Science and Technology (Grant No. 202403AC100010, awarded to WH-L). The funders had no role in study design, data collection and analysis, decision to publish, or preparation of the manuscript.

**Competing interests:** The authors have declared that no competing interests exist.

prolongation without statistical significance observed. Under the same conditions, theses drugs showed no effects on king cobra venoms and significantly prompted the death of the tested animals. Present investigation provides a scientific basis for the treatment of *B. multicinctus* bites in remote mountainous areas or rural regions of southern China. Upon confirming victims were bitten by *B. multicinctus*, administering atropine sulfate or neostigmine methylsulfate as emergency treatment might provide supplementary benefits for subsequent care. Importantly, the administration of clinic used drugs does not interfere with the efficacy of *B. multicinctus* antivenin in later treating stages.

## Author summary

*B. multicinctus* envenomation represents a critical public health issue, primarily concentrated in China and Southeast Asian countries. This study evaluated the protective effects of clinically standard anticholinergic and anticholinesterase drugs against bungarotoxins and neurotoxin-rich crude snake venoms. Neostigmine methylsulfate and atropine sulfate not only demonstrated significantly protective effects against combinations of three-finger toxins, β-bungarotoxin+α-bungarotoxin, and β-bungarotoxin+γ-bungarotoxin in *B. multicinctus* venom, but also provided moderate protective effects against *B. multicinctus* crude venom. Notably, these two drugs did not interfere with the subsequent administration of antivenin. These findings establish a scientific basis for treating *B. multicinctus* envenomation in remote regions, with atropine sulfate is recommended for use in victims unable to access antivenin as a first-aid remedy.

## Introduction

According to the World Health Organization (WHO) reports, approximately 5 million people are bitten by snakes globally each year, leading to 2.7 million cases of envenomation, 81,000–138,000 deaths, and 400,000 disabilities [1,2]. China is rich in snake biodiversity, with over 600 snake species recorded, including over 100 venomous species distributed widely across the country. Venomous snakebite remains a critical public health issue in China, where the current mortality rate reaches 5%-10% and disability rates remain high [3,4]. According to 2024 published the Chinese guideline for management of snakebites, venomous snakebites occurred in China around 250,000–280,000 cases each year. Notably, the snakebite mortality is mainly caused by neurotoxin-rich Elapidae venoms [5]. Despite these statistics, snakebite treatment is often overlooked, and standardized, reliable diagnostic methods are lacking [6]. Antivenin remains the primary treatment for snakebite envenomation to reduces morbidity and mortality but it has significant limitations in China: Firstly, only four kinds of monovalent antivenins are commercially available, and many clinically relevant antivenin are unavailable. Secondly, cost and cold chain transportation challenges

lead to insufficient antivenin supply in certain regions [7]. Thirdly, side effects of antivenin can complicate treatment [8–10]. Lastly, snakebite high-incidence areas are difficult to maintain long-term stability and efficacy of antivenin [11].

The snake species of *B. multicinctus* is widely distributed in south of China and neighboring countries such as Vietnam, Pakistan, and Myanmar [12–14]. This species possesses extremely potent venoms capable of rapidly inducing respiratory paralysis and death [15]. The venom composition is complex, primarily consisting of diverse polypeptides and proteins [16,17], which act through distinct mechanisms on the nervous and muscular systems, leading to symptoms such as muscle weakness, dyspnea, ptosis, and even fatal outcomes [18,19]. Key toxins in *B. multicinctus* venom include: α-bungarotoxin, γ-bungarotoxin, β-bungarotoxin and κ-bungarotoxin [16,20]. α-bungarotoxin is a post-synaptic neurotoxin that competitively inhibits acetylcholine binding to receptors, blocking neuromuscular signal transmission [12,21,22]. β-bungarotoxin is a presynaptic neurotoxin with an intraperitoneal $LD_{50}$ of 0.004 μg/g and represented as the most lethal component in the venom [16,23]. It disrupts neuromuscular junctions and induces cellular damage, altering acetylcholine levels in synaptic clefts [24,25]. At present, the γ-bungarotoxin is regarded as the most toxic three-finger toxin in so far identified snake venom three-finger toxins [26]. It can react with M2 acetylcholine receptors as well as integrin alpha 5 to damage blood-brain barrier (BBB) [20,26], but its precise mechanism remains unclear [27,28]. Notably, the primary sequence of γ-bungarotoxin contains an Arg-Gly-Asp (RGD) motif, conferring multifunctional bioactivity with therapeutic potential [16,29,30]. κ-bungarotoxin is a less-studied post-synaptic neurotoxin that potently blocks neuromuscular signaling [31].

Given the complicated acting mechanisms of different bungarotoxins, this study evaluated the protective effects of clinically used drugs targeting acetylcholine receptors (bethanechol chloride, pilocarpine hydrochloride, scopolamine, atropine sulfate) and acetylcholinesterase (neostigmine methylsulfate) against different bungarotoxins and other neurotoxin-rich venoms [32,33]. Our previous laboratory studies on *B. multicinctus* envenomation showed that antivenin effectively neutralizes free β-bungarotoxin in blood circulation but has limited efficacy against three-finger neurotoxins (α-bungarotoxin, γ-bungarotoxin, κ-bungarotoxin) [16]. In remote mountainous/rural regions of China and neighboring countries, antivenin may be unavailable following *B. multicinctus* bites and might delay effective treatment. We therefore investigated whether early administration of clinically standard anticholinergic/anticholinesterase drugs could protect patients or prolong survival during the pre-antivenin period, without interfering with subsequent antivenin efficacy. Future research should focus on developing safer, more effective antivenin alternatives and unraveling the toxicological mechanisms of *B. multicinctus* venom to establish standardized, reliable diagnostic and therapeutic protocols for this kind of snakebite management.

## Experimental materials and reagents

### Ethical statement

All experiments on animals meet the requirements of National Institutes of Health guide for the care and use of Laboratory animals (NIH Publications No. 8023) and has been reviewed and approved by Animal Care and Use Committee of Kunming Institute of Zoology, Chinese Academy of Sciences (Approval ID: SMKX-2017023). Kunming mice (20±3 g) were provided and bred by the Animal Center of the Kunming Institute of Zoology, Chinese Academy of Sciences.

### Venoms, drugs and antivenins

Pooled *B. multicinctus* venom was purchased from a snake farm in Zhejiang Province, China. *N. atra* and *O. hannah* were stored samples from the Functional Proteomics of Natural Medicines Group at the Kunming Institute of Zoology, the Chinese Academy of Sciences. All the used venoms are collected from corresponding adult snakes which are born and grow in snake farms. Antivenin against *B. multicinctus* was purchased from Shanghai Sailun Biotechnology Company (Batch No. 20210301, 10000 U/ml). Neostigmine methylsulfate, scopolamine, atropine sulfate, bethanechol chloride and pilocarpine hydrochloride are all produced by the American Sellk Company. Details of the drugs are as follows: Scopolamine (Batch No. S5873, 99.99%), neostigmine methylsulfate (Batch No. S5886, 99.99%), bethanechol Chloride (Batch No.

S2455, 99.99%), pilocarpine hydrochloride (Batch No. S1608, 99.99%), and atropine sulfate (Batch No. S5493, 99.99%). α-bungarotoxin, β-bungarotoxin and γ-bungarotoxin were isolated from *B. multicinctus* crude venom according to our previously described methods and stored for use [16]. All snake venoms, various bungarotoxins, and drugs were stored at -20°C, while the *B. multicinctus* Antivenin was maintained at 4°C.

## Experimental methods

### Protective effects of pre-administered anticholinergic and anticholinesterase drugs against different snake venoms

The present study aimed to investigate the protective effects of anticholinergic and anticholinesterase drugs against different bungarotoxins and neurotoxin-rich snake venoms. Prior to this experiment, neostigmine methylsulfate, bethanechol chloride, and pilocarpine hydrochloride were administered via subcutaneous injection (SC, the inferior dorsal region of mice, the same applies hereinafter), while scopolamine was administered via intraperitoneal injection (IP, the inferior abdominal region at the midpoint between the hindlimbs and ventral midline, the same applies hereinafter). After allowing 10 minutes for the drug to take effect, mice received IP injection of relevant snake venoms. Each experimental group independently utilized six mice to record survival times. Dosage calculations: α-bungarotoxin and γ-bungarotoxin: $2 \times LD_{50}$, the selected dosage is benefit to evaluate the effects of the drugs because the higher dosage make the tested animals dead quickly [16]. β-bungarotoxin, *B. multicinctus* crude venom, *N.atra* crude venom and *O. hannah* crude venom: $3 \times LD_{50}$. *N. atra* crude venom $LD_{50}$: 0.5 mg/kg. *O. hannah* crude venom $LD_{50}$: 1 mg/kg. Neostigmine methylsulfate: 0.1 mg/kg. Scopolamine: 10 mg/kg. Bethanechol chloride and pilocarpine hydrochloride: 1 mg/kg. Protective effects were assessed by monitoring survival outcomes.

### Protective effects of anticholinergic and anticholinesterase drugs against different pre-administered bungarotoxins

In vivo venom-induced lethality protection assays were mainly according to the methods described by Ahmadi et al [34]. To mimic real-world scenarios of *B. multicinctus* envenomation, mice were pre-injected with different bungarotoxins via IP injection in the present study. Ten minutes later, neostigmine methylsulfate or atropine sulfate was administered via either SC or intravenous (IV) injection, while scopolamine hydrobromide was administered via IP injection. Each experimental group independently utilized six mice to record survival times. Dosages were as follows: α-bungarotoxin, γ-bungarotoxin, neostigmine methylsulfate and scopolamine were administered at the same doses as in Section: Protective Effects of Pre-Administered Anticholinergic and Anticholinesterase Drugs against Different Snake Venoms. Atropine sulfate: 1.5 mg/kg. The selected time intervals between drug and venom injections is based on the death time of only venom group and the time intervals of the drugs after venom injections were approximately 1/4 of the death time which might be mimic clinical relevance of the snake-bite situations.

### Protective effects of anticholinergic and anticholinesterase drugs combined with antivenin against different bungarotoxins

This study was designed to better replicate scenarios in which antivenin specific to *B. multicinctus* is unavailable within a clinically relevant timeframe following bites by this species in remote areas of southern China. To evaluate whether pre-administration of neostigmine methylsulfate and atropine sulfate could provide protective effects before antivenin injection, mice were pre-injected with different bungarotoxins via IP injection. After 3 minutes, drugs (neostigmine methylsulfate, atropine sulfate) were administered by SC routines, respectively. Antivenin was then IV injected after 6 minutes. Each experimental group independently utilized six mice to record survival times. Dosage calculations: $ED_{50}$ (median effective dose) was calculated using the Spearman-Karber method ($ED_{50}$: 17.68 mg/kg) with a $2 \times ED_{50}$ dosage applied.

α-bungarotoxin and γ-bungarotoxin: 2×LD$_{50}$. β-bungarotoxin, *B. multicinctus* crude venom, neostigmine methylsulfate, and atropine sulfate: Dosages matched those in Section: Protective Effects of Anticholinergic and Anticholinesterase Drugs Against Different Pre-Administered Bungarotoxins.

### Randomization

One day prior to the start of the experiment, all mice were weighed and ranked in ascending order of body weight. According to the experimental design, the animals were then randomly allocated to the negative control group and various drug-protected groups, ensuring an even distribution of individual weights across groups and minimizing selection bias.

### Effects of individual drugs on mice

Preliminary experiments were conducted involving pharmacological control groups. Specifically, a saline control group and various drug treatment groups were established, Neostigmine methylsulfate, atropine sulfate, bethanechol chloride, and pilocarpine hydrochloride were administered to mice via subcutaneous injection; neostigmine methylsulfate and atropine sulfate via intravenous injection; and scopolamine via intraperitoneal injection. The survival time of the mice was observed and recorded to evaluate the effects of these agents. A total of 6 mice per group were used to record survival times.

### Drug dosage calculation formula

Based on mouse-to-human dose conversion [Animal Dose (mg/kg) = Human Dose (mg/kg) × (Human Body Surface Area/ Animal Body Surface Area) × (Animal Body Weight/ Human Body Weight)}. Where the body surface area (BSA) is calculated by the formula: $S = k × W^{2/3}$ (k: species constant, human = 10.6, mouse = 9.1, W: body weight, kg. The formula is based on a standard body weight of 70 kg for an adult human and 20 g for a mouse) as indicated in S1 Table [35,36].

### Statistical analysis

Sample size was determined based on effect sizes reported in published studies with similar experimental designs, with a minimum of $n = 6$ mice per group to ensure adequate statistical power. Individual data points are displayed in scatter plots. Measurement data are expressed as mean ± standard deviation (SD). All statistical analyses were performed using GraphPad Prism software (version 10.1). Normality of data distribution was assessed using the Shapiro-Wilk test, and homogeneity of variances was evaluated with Levene's test. For comparisons between two independent groups: Unpaired Student's *t*-test was applied when data met assumptions of normality and equal variance. Welch's *t*-test correction was used when variances were significantly unequal. For comparisons among three or more independent groups: One-way analysis of variance (ANOVA) was conducted if normality and homogeneity of variances were satisfied. Tukey's honestly significant difference (HSD) post hoc test was performed for multiple pairwise comparisons only when the overall ANOVA yielded a significant result ($p < 0.05$). All measurement data are reported with 95% confidence intervals (95% CIs) to assess both the precision of effect estimates and their statistical significance. Statistical significance thresholds were defined as follows: ns $p > 0.05$, * $p < 0.05$, ** $p < 0.01$, *** $p < 0.001$, **** $p < 0.0001$. These significance levels apply to all relevant statistical outputs (*t*-tests, ANOVA main effects, and post hoc comparisons).

## Experimental results

### Protective effects of pre-administered anticholinergic and anticholinesterase drugs against different snake venoms

This study investigated the protective effects and underlying mechanisms of anticholinergic and anticholinesterase drugs against different bungarotoxins and other neurotoxin-rich snake venoms, yielding the following key results (Fig 1): For γ-bungarotoxin: Pre-administration of scopolamine and neostigmine methylsulfate conferred highly significant protective

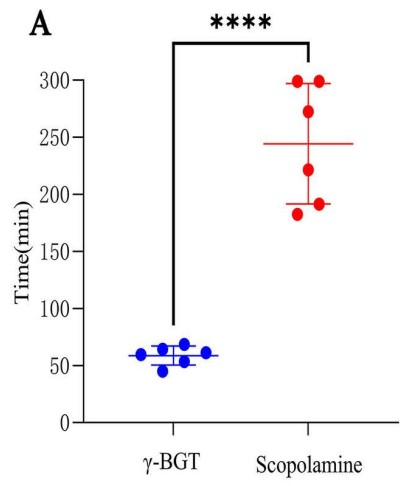

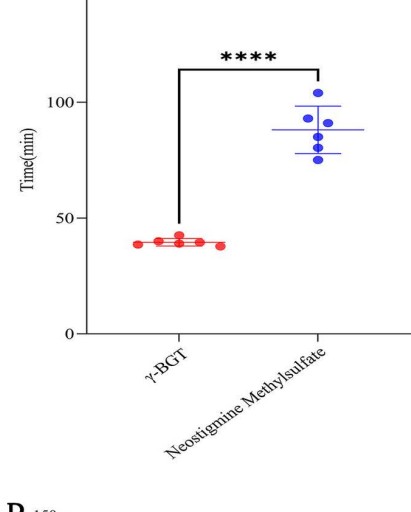

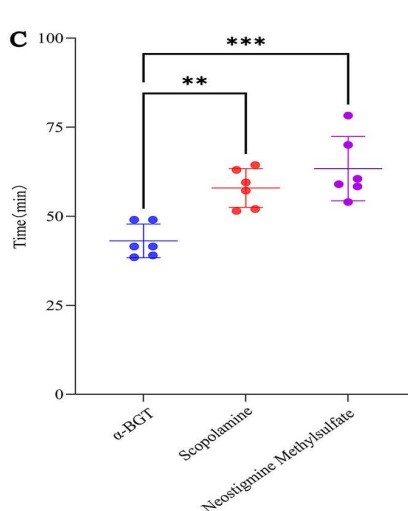

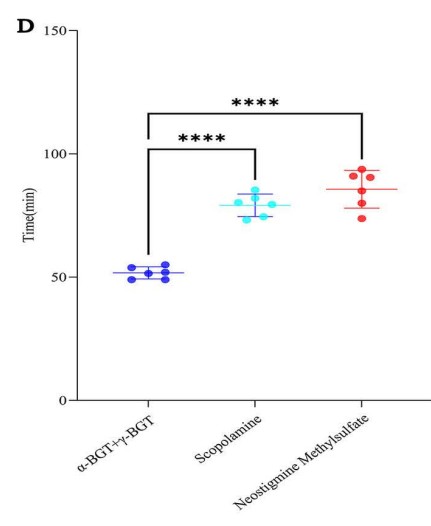

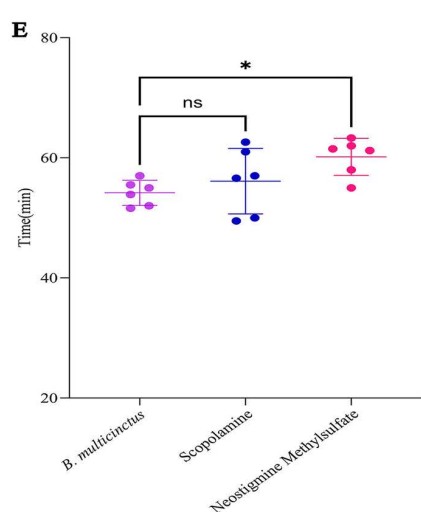

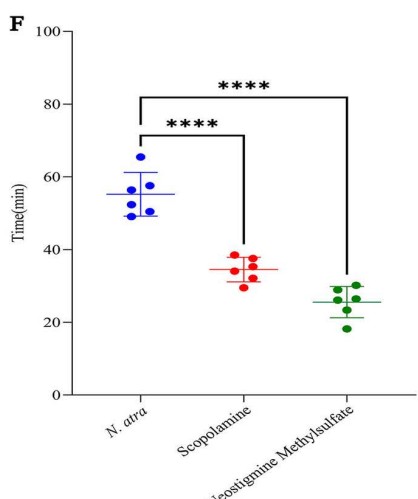

**Fig 1. Pre-administration of Scopolamine and Neostigmine Methylsulfate Demonstrated Markedly Protective Effects Against Three-Finger Toxins from *B. multicinctus* Venom.** A. Protective effects of pre-administration with scopolamine on γ-bungarotoxin. B. Protective effects of pre-administration with neostigmine methylsulfate on γ-bungarotoxin. C. Protective effects of pre-administration with scopolamine or neostigmine methylsulfate on α-bungarotoxin. D. Protective effects of pre-administration with scopolamine or neostigmine methylsulfate on α-bungarotoxin + γ-bungarotoxin combination. E. Protective effects of pre-administration with scopolamine or neostigmine methylsulfate on *B. multicinctus* crude venom. F. Protective effects of pre-administration with scopolamine or neostigmine methylsulfate on *N. atra* crude venom. Results were expressed as mean ± SD; n = 6 per group. Significance levels were indicated as follows: ns $p > 0.05$, * $p < 0.05$, ** $p < 0.01$, *** $p < 0.001$, **** $p < 0.0001$.

effects (Fig 1A & 1B, $p < 0.0001$), whereas bethanechol chloride and pilocarpine hydrochloride showed no protective activity (S1A Fig). For α-bungarotoxin: Both scopolamine and neostigmine methylsulfate demonstrated significant protective effects (Fig 1C, $p < 0.0001$), with neostigmine methylsulfate yielding superior outcomes. For β-bungarotoxin and *O. hannah* crude venom: Neither scopolamine nor neostigmine methylsulfate provided effective protection (S1B & S1C Fig, $p > 0.05$). For the combination of α- and γ-bungarotoxins, both scopolamine and neostigmine methylsulfate exerted strong protective effects (Fig 1D, $p < 0.0001$). For *B. multicinctus* crude venom: Neostigmine methylsulfate significantly prolonged survival (Fig 1E, $p < 0.05$), while scopolamine marginally increased survival time without statistical significance (Fig 1E, $p > 0.05$). For *N. atra* crude venom: Scopolamine and neostigmine methylsulfate significantly accelerated mortality (Fig 1F, $p < 0.0001$). The survival times of the experimental animals were recorded and provided in S2 Table.

### Protective effects of anticholinergic and anticholinesterase drugs against different pre-administered bungarotoxins

Realistic envenomation scenarios for *B. multicinctus* were carried out mimicly with key representations shown in Fig 2. For the pre-administration of γ-bungarotoxin + α-bungarotoxin: Both neostigmine methylsulfate and atropine sulfate provided significant protective effects, with neostigmine methylsulfate showing more pronounced outcomes (Fig 2A & 2B, $p < 0.01$). We compared IV versus SC administration routes to evaluate the superior efficacy of neostigmine methylsulfate and atropine sulfate against γ-bungarotoxin, with IV delivery initiated at 10 minutes post-exposure. Results showed that atropine sulfate exhibited significantly better protective effects against γ-bungarotoxin than that of SC administration (Fig 2C, $p < 0.0001$). For the pre-administration of α-bungarotoxin+γ-bungarotoxin combination, all of the tested scopolamine, neostigmine methylsulfate and atropine sulfate exerted significant protective effects. Among these, neostigmine methylsulfate and atropine sulfate demonstrated more pronounced protective efficacy (Fig 2D, $p < 0.0001$). For the pre-administration of γ-bungarotoxin, additional experiments were conducted to investigate the dose-response relationships. Double-standard doses of neostigmine methylsulfate and atropine sulfate were administered intravenously (IV) at 10 minutes. Results showed: Atropine sulfate exerted no significant effect (S2A Fig, $p > 0.05$), while neostigmine methylsulfate accelerated mortality (S2A Fig, $p < 0.0001$). Subsequent dose-escalation studies revealed that IV neostigmine methylsulfate used at doses exceeding 0.1 mg/kg caused a dose-dependent decline in protective efficacy, eventually transitioning to lethal outcomes. To simulate situations involving prolonged treatment delay and when animals near death after γ-bungarotoxin administration, drugs were administered IV 20-minute later. Our findings demonstrated that neostigmine methylsulfate retained significant protective effects (S2B Fig, $p < 0.01$). Atropine sulfate marginally prolonged survival but lacked statistical significance (S2B Fig, $p > 0.05$). In a simulated scenario of actual *B. multicinctus* envenomation, mice were first pre-injected with *B. multicinctus* crude venom, followed by subcutaneous injection of Neostigmine Methylsulfate or Atropine Sulfate respectively. The results showed that both drugs prolonged the survival time of mice to a certain extent, but the difference was not statistically significant (Fig 2E, $p > 0.05$). In the mouse model pre-injected with *N. atra* crude venom, Neostigmine Methylsulfate not only failed to provide protective effects but also significantly accelerated the death of the animals (Fig 2F, $p < 0.0001$)., while Atropine Sulfate exerted no significant effect (Fig 2F, $p > 0.05$)."

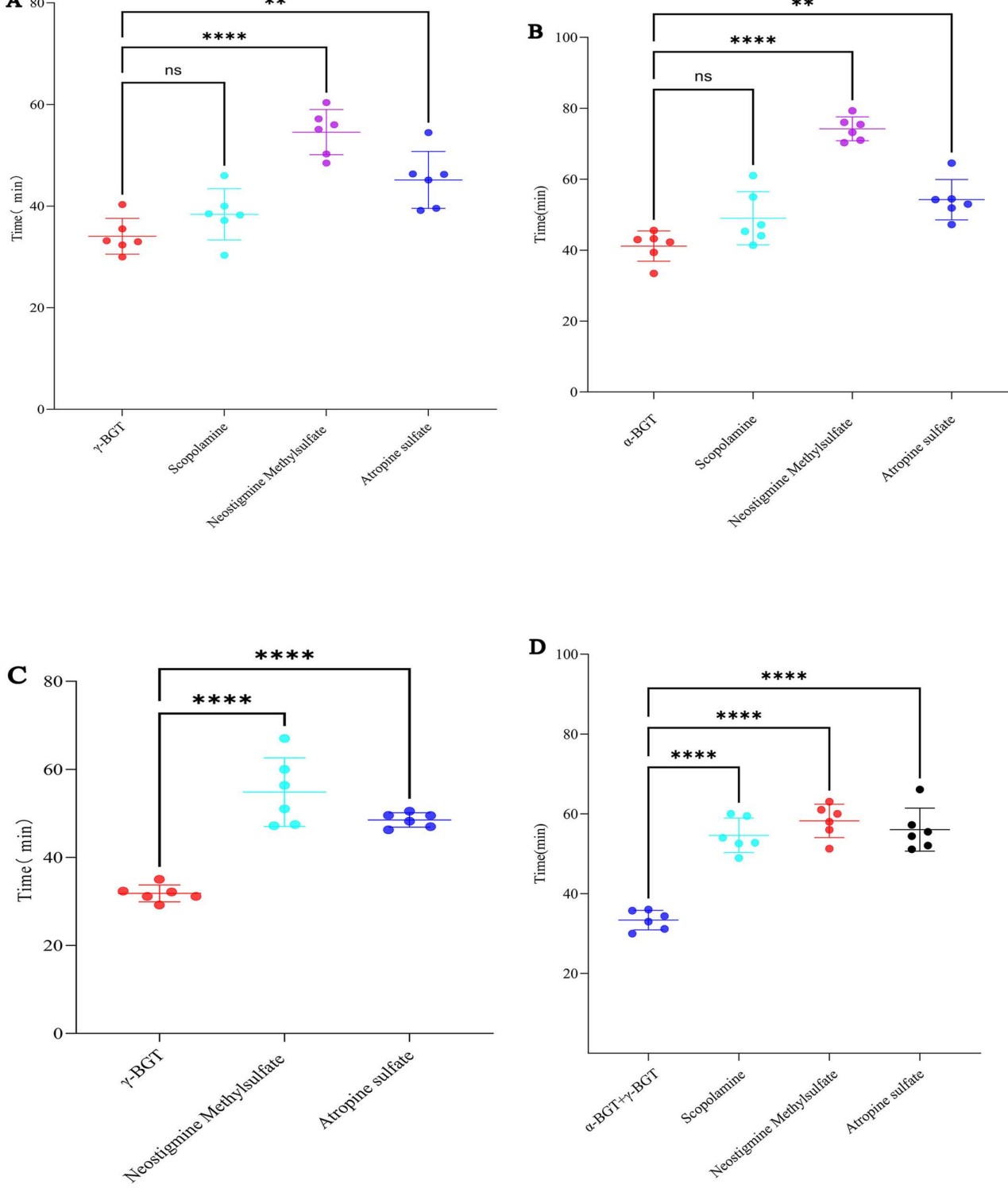

**Fig 2. Neostigmine Methylsulfate and Atropine Sulfate Also Demonstrated Markedly Protective Effects Against Pre-injected Three-Finger Toxins from *B. multicinctus* Venom.** A. Protective effects of scopolamine, neostigmine methylsulfate, and atropine sulfate on pre-administration of γ-bungarotoxin. B. Protective effects of scopolamine, neostigmine methylsulfate, and atropine sulfate on pre-administration of α-bungarotoxin. C. Protective effects of neostigmine methylsulfate and atropine sulfate administered IV at 10 minutes on pre-administration of γ-bungarotoxin. D. Protective effects

of scopolamine, neostigmine methylsulfate, and atropine sulfate on pre-administration of α-bungarotoxin + γ-bungarotoxin combination. E. Protective effects of neostigmine methylsulfate and atropine sulfate on pre-administration of *B. multicinctus* crude venom. F. Protective effects of neostigmine methylsulfate and atropine sulfate on pre-administration of *N. atra* crude venom. Results were expressed as mean ± SD; n = 6 per group. Significance levels were indicated as follows: ns $p > 0.05$, * $p < 0.05$, ** $p < 0.01$, *** $p < 0.001$, **** $p < 0.0001$.

## Protective effects of anticholinergic and anticholinesterase drugs combined with antivenin against different bungarotoxins

In certain situations where antivenin against *B. multicinctus* is unavailable timely within a short time after snakebites in remote mountainous or rural areas of southern China, the first-aid remedy is very important to save life. Fig 3 illustrates that: for the pre-administration of β-bungarotoxin+γ-bungarotoxin combination, both neostigmine methylsulfate and atropine sulfate exerted highly significant protective effects, with atropine sulfate demonstrating superior outcomes (Fig 3A, $p < 0.01$). For the pre-administration of β-bungarotoxin+α-bungarotoxin, the protective effects of neostigmine methylsulfate and atropine sulfate paralleled those observed in the β + γ-bungarotoxin group (Fig 3C, $p < 0.01$). In mice receiving antivenin against *B. multicinctus*, pre-treatment with neostigmine methylsulfate/atropine sulfate+antivenin survived significantly longer compared to antivenin-only controls. The atropine sulfate+antivenin combination yielded the most pronounced protective effects (Fig 3B & 3D, $p < 0.01$). For *B. multicinctus* crude venom, combined administration of neostigmine methylsulfate/atropine sulfate+antivenin marginally prolonged survival but lacked statistical significance (Fig 3E, $p > 0.05$).

## Conclusions and discussion

Worldwide, over 600 venomous snake species have been identified. In mainland China, *B. multicinctus* ranks among the most venomous and lethal snake species. With a single venom injection quantity of about 4.6 mg dry weight, it far surpasses the lethal dose for humans. Reported cases of envenomation by *B. multicinctus* have demonstrated mortality rates as high as 23% [37]. The situation of *B. multicinctus* bites is extremely dangerous. If appropriate treatment is not given promptly after a bite, the mortality rate can exceed 50%. Even when medical help is provided, some patients still suffer long-term consequences. Prolonged poisoning and severe nerve damage can lead to problems such as limb impairment and other post-bite complications [38]. Given these facts, research on *B. multicinctus* venom and the treatment of its bites is of utmost importance. It holds significant clinical value, offering essential insights for both emergency response and long-term management strategies.

This study evaluated the protective effects of clinically standard anticholinergic and anticholinesterase drugs—including scopolamine, neostigmine methylsulfate, bethanechol chloride, pilocarpine hydrochloride, and atropine sulfate—against different components of *B. multicinctus* venom and neurotoxin-rich snake venoms (*B. multicinctus*, *N. atra* and *O. hannah*). Firstly, drugs were pre-administered to mice, followed by venom challenge after therapeutic concentrations were achieved. This evaluated their protective efficacy and underlying mechanisms against bungarotoxins and neurotoxin-rich snake venoms. Secondly, mice were pre-injected with specific bungarotoxins to simulate real-world envenomation scenarios. Drug efficacy against these pre-injected toxins was then analyzed. Thirdly, mimicking scenarios where antivenin is unavailable in remote southern Chinese rural/mountainous areas, drugs were administered prior to antivenin following venom challenge. This assessed whether drugs retained protective effects when antivenin treatment was delayed.

Our investigation demonstrated that pre-administration of scopolamine and neostigmine methylsulfate exerted highly significant protective effects against the α-bungarotoxin, γ-bungarotoxin, and their combination, respectively. Neostigmine methylsulfate also showed significant preventive effects against *B. multicinctus* crude venom. These findings suggest antagonistic interactions between these drugs and α/γ-bungarotoxins, particularly providing insights into γ-bungarotoxin binding sites. Notably, scopolamine and neostigmine methylsulfate accelerated mortality in mice challenged with *N. atra* venom. This observation hypothesizes synergistic pro-death effects between the drugs and specific *N.atra* venom

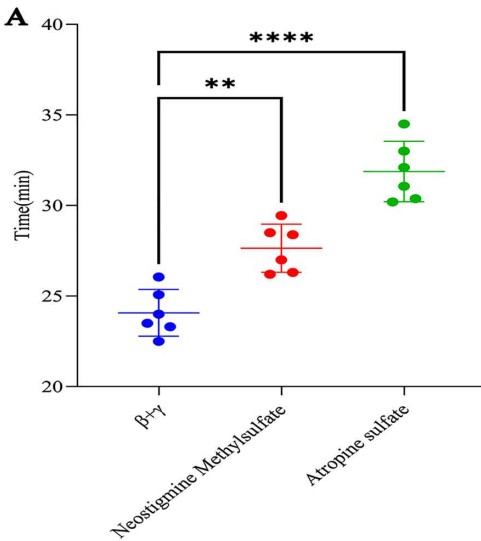

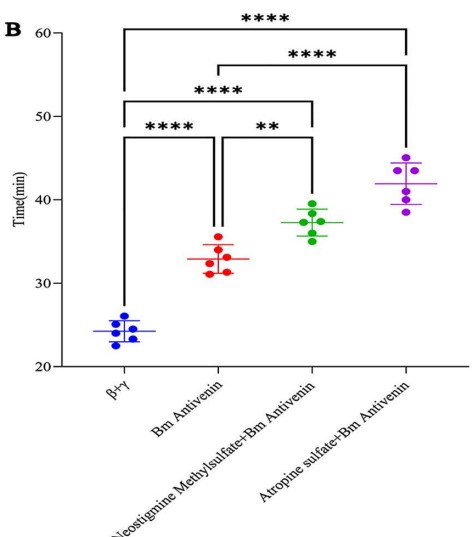

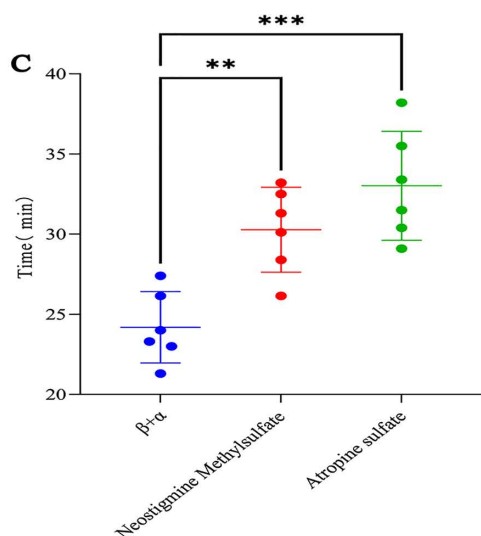

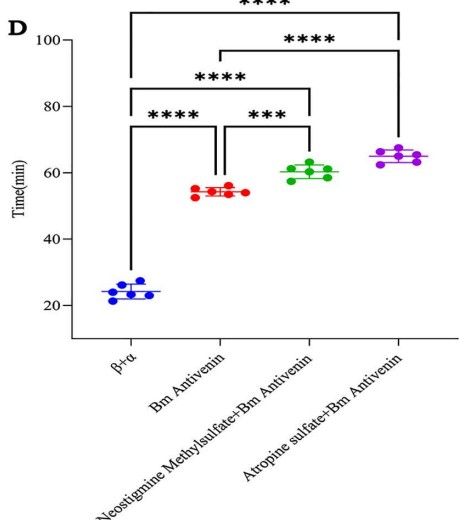

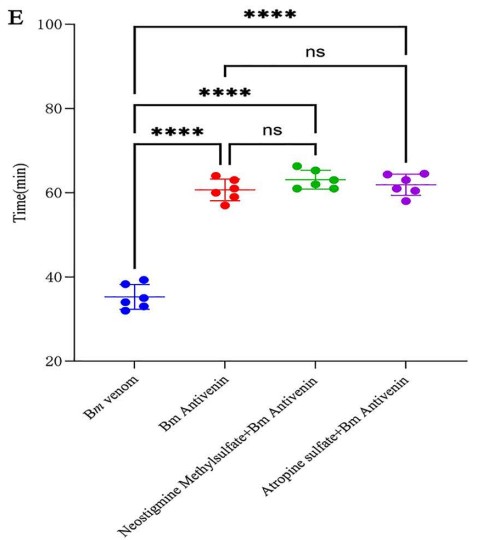

**Fig 3. Neostigmine Methylsulfate and Atropine Sulfate Exert Markedly Protective Effects Against the β-Bungarotoxin+Three-Finger Toxin Complex in *B. multicinctus* Venom.** A. Protective effects of neostigmine methylsulfate and atropine sulfate on pre-administration of β-bungarotoxin+γ-bungarotoxin. B. Protective effects of neostigmine methylsulfate/atropine sulfate combined with *B. multicinctus* antivenin against β-bungarotoxin+γ-bungarotoxin. C. Protective effects of neostigmine methylsulfate and atropine sulfate on pre-administration of β-bungarotoxin+α-bungarotoxin. D. Protective effects of neostigmine methylsulfate and atropine sulfate combined with *B. multicinctus* antivenin against β-bungarotoxin+α-bungarotoxin. E. Protective effects of neostigmine methylsulfate and atropine sulfate combined with *B. multicinctus* antivenin against *B. multicinctus* crude venom. Results were expressed as mean±SD; n=6 per group, Significance levels were indicated as follows: ns $p > 0.05$, * $p < 0.05$, ** $p < 0.01$, *** $p < 0.001$, **** $p < 0.0001$.

components, underscoring the critical importance of accurate snakebite diagnosis. Secondly, by pre-administering bungarotoxins to simulate real-world envenomation scenarios, this study demonstrated that neostigmine methylsulfate and atropine sulfate exerted highly significant protective effects against α-bungarotoxin, γ-bungarotoxin, and their combination. These findings hypothesize overlapping binding sites between the drugs and α/γ-bungarotoxins, with neostigmine methylsulfate and atropine sulfate exhibiting superior binding affinity. Notably, intravenous administration of neostigmine methylsulfate and atropine sulfate revealed that atropine sulfate conferred more pronounced protection at standard doses. Subsequent experiments showed that intravenous administration of doubling neostigmine methylsulfate dosage paradoxically accelerated mortality, whereas standard-dose neostigmine methylsulfate administered at the terminal stage (20 minutes post-envenomation) exerted significant protective effects. This biphasic response may relate to rapid drug bioavailability via intravenous injection, underscoring the critical need for strict dosage control, particularly during intravenous delivery. Finally, when neostigmine methylsulfate and atropine sulfate were administered prior to antivenin injection following bungarotoxins challenge, both drugs exerted highly significant protective effects against β-bungarotoxin+γ-bungarotoxin and β-bungarotoxin+α-bungarotoxin combinations, with atropine sulfate demonstrating superior outcomes. In mice receiving antivenin against *B. multicinctus*, pre-treatment with neostigmine methylsulfate/atropine sulfate+antivenin survived significantly longer compared to antivenin-only controls. The atropine sulfate+antivenin combination yielded the most pronounced protective effects. For *B. multicinctus* crude venom, combined administration of neostigmine methylsulfate/atropine sulfate+antivenin marginally prolonged survival but lacked statistical significance. This may be attributed to high β-bungarotoxin contained in *B. multicinctus* crude venom, potentially overwhelming drug efficacy. Given the $PLA_2$ activity of β-bungarotoxin, future studies could evaluate $PLA_2$ inhibitors to counteract its toxicity. Additionally, optimizing dosages of neostigmine methylsulfate and atropine sulfate in *B. multicinctus* crude venom models may enhance protective efficacy. Although the neurotoxic symptoms were mainly attributed to β-bungarotoxin, the effects of other three-finger bungarotoxins contained in the venom should have their roles contributed in neurotoxic symptoms. Thus, neostigmine methylsulfate and atropine sulfate might have their value in certain conditions where victims bitten by *B. multicinctus* might benefit by administration these drugs before antivenin available.

Recently, the Chinese guideline for management of snakebites also recommended to use anticholinesterase drugs such as neostigmine or pyridostigmine to reduce the hydrolysis of acetylcholine in the synaptic gap and exert a complete cholinesteroid effect in the reversal of some types of neurotoxin-induced myocardial paralysis. Neostigmine or pyridostigmine 0.02 mg/kg (0.04 mg/kg in children) is administered intramuscularly, with 0.5–2.5 mg repeated for 1–3 h if necessary, with the total daily dose not exceeding 10 mg. Because of the risk of increased airway secretions with neostigmine, 0.6 mg of atropine sulfate (50 μg/kg in children) may be administered intravenously prior to the medication. Recommendation 21 of the Chinese guideline pointed out that anticholinesterase drugs are indicated only for patients with neurotoxic snakebites in whom antivenin is not effective (Evidence level B, Recommendation II). It should be noted that such medications should not delay the administration of antivenin and necessary tracheal intubation [5,38,39]. Previously, one patient out of eight victims envenoming by *B. multicinctus* in Myanmar was received anticholinesterase injection but failed to improve neurotoxic symptoms [38]. However, the reported intravenous administration of prostigmine (0.5 mg) and atropine (0.2 mg) failed to improve the neurological symptoms. The ineffectiveness of the drugs might be caused by very late administration

of the drugs because the patient was received *N. kaouthia* antivenin (no specific antivenin was available) and showed ptosis on arrival the hospital (14 h after the bite).

In summary, present investigation provides a scientific basis for the treatment of *B. multicinctus* envenomation, particularly in rural/remote mountainous regions of southern China where antivenin against *B. multicinctus* may be unavailable in the acute phase. Our findings highlight atropine sulfate as the most effective adjuvant therapy. Administration of atropine sulfate following confirmed envenoming by *B. multicinctus* bites can extend the window for definitive treatment, significantly improving outcomes without interfering with subsequent antivenin efficacy. These results also offer valuable insights for managing severe *B. multicinctus* envenomation cases. Notably, the finding that scopolamine and neostigmine methylsulfate accelerated mortality rates in mice challenged with *N.atra* crude venom underscores the critical importance of accurate snakebite diagnosis. Future research should focus on developing safer and more effective antivenin alternatives while elucidating the toxicity mechanisms of *B. multicinctus* venom. These efforts will help establish standardized, evidence-based protocols for the diagnosis and treatment of *B. multicinctus* envenomation.

Critical limitations: Animal experimental results indicated that neostigmine methylsulfate and atropine sulfate might be used for treating *B. multicinctus* victims but not for cobra and king cobra envenomation.

## Supporting information

**S1 Fig. Effects of single drugs (pre-injected) on mouse survival.** A: Protective effects of Bethanechol Chloride and Pilocarpine Hydrochloride against γ-bungarotoxin. Mice were pre-injected with the two drugs prior to γ-bungarotoxin administration; neither drug exerted a significant protective effect (p > 0.05, n = 6 per group). B: Protective effects of scopolamine and neostigmine methylsulfate against β-bungarotoxin. Mice were pre-injected with the two drugs prior to β-bungarotoxin administration; neither drug exerted a significant protective effect (p > 0.05, n = 6 per group). C: Protective effects of scopolamine and neostigmine methylsulfate against *Ophiophagus hannah* venom. Mice were pre-injected with the two drugs prior to *O. hannah* venom administration; neither drug exerted an effective protective effect (p > 0.05, n = 6 per group).
(TIFF)

**S2 Fig. Effects of single drugs on mouse survival (different injection doses and timings).** A: Protective effects of double-dose neostigmine methylsulfate and atropine sulfate against γ-bungarotoxin. After γ-bungarotoxin administration to mice, intravenous injection of double-dose atropine sulfate showed no significant effect (p > 0.05), while double-dose neostigmine methylsulfate accelerated mortality (p < 0.0001, n = 6 per group). B: Protective effects of neostigmine methylsulfate and atropine sulfate administered at 20 minutes post γ-bungarotoxin injection. Mice were administered the drugs via injection at 20 minutes after γ-bungarotoxin exposure: neostigmine methylsulfate still exerted a significant protective effect (p < 0.01), while atropine sulfate only slightly prolonged survival (p > 0.05, n = 6 per group).
(TIFF)

**S1 Table. Mouse-to-Human Drug Dose Conversion.** This table presents the conversion relationship between mouse and human drug doses.
(TIF)

**S2 Table. Control Groups for Different Drugs, Administration Routes and Administration Times.** This table summarizes the control groups corresponding to different drugs, injection routes, and injection timings.
(PNG)

## Author contributions

**Conceptualization:** Guowen Zhang, Mengqi He, Wen-Hui Lee.

**Data curation:** Guowen Zhang.

**Formal analysis:** Guowen Zhang, Mengqi He.

**Funding acquisition:** Wen-Hui Lee.

**Investigation:** Guowen Zhang, Mengqi He, Tongyi Sun, Wen-Hui Lee.

**Methodology:** Guowen Zhang, Mengqi He.

**Resources:** Wen-Hui Lee.

**Supervision:** Tongyi Sun, Wen-Hui Lee.

**Writing – original draft:** Guowen Zhang.

**Writing – review & editing:** Guowen Zhang, Wen-Hui Lee.

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
