## [Decision Letter · Decision Letter 0]

20 Jun 2025

Protective Effects of Clinical Anticholinergic and Anticholinesterase Agents Against Bungarus multicinctus Venom and Neurotoxin-Rich Snake Venoms

Dear Dr. Lee,

Thank you for submitting your manuscript to PLOS Neglected Tropical Diseases. After careful consideration, we feel that it has merit but does not fully meet PLOS Neglected Tropical Diseases's publication criteria as it currently stands. Therefore, we invite you to submit a revised version of the manuscript that addresses the points raised during the review process.

Please submit your revised manuscript within 60 days Aug 19 2025 11:59PM. If you will need more time than this to complete your revisions, please reply to this message or contact the journal office at plosntds@plos.org. Please include the following items when submitting your revised manuscript:

We look forward to receiving your revised manuscript.

Kind regards,

Wuelton Monteiro, Ph.D.

Section Editor

Wuelton Monteiro

Section Editor

Shaden Kamhawi

co-Editor-in-Chief

Paul Brindley

co-Editor-in-Chief

**Journal Requirements:**

At this stage, the following Authors/Authors require contributions: Guowen Zhang, Mengqi He, Tongyi Sun, and Wen-Hui Lee. Please ensure that the full contributions of each author are acknowledged in the "Add/Edit/Remove Authors" section of our submission form.

4) Please amend your detailed Financial Disclosure statement. This is published with the article. It must therefore be completed in full sentences and contain the exact wording you wish to be published. Please ensure that the funders and grant numbers match between the Financial Disclosure field and the Funding Information tab in your submission form. Note that the funders must be provided in the same order in both places as well.

**Reviewers' Comments:**

Reviewer's Responses to Questions

**Key Review Criteria Required for Acceptance?**

**Methods**

-Are the objectives of the study clearly articulated with a clear testable hypothesis stated?

-Is the study design appropriate to address the stated objectives?

-Is the population clearly described and appropriate for the hypothesis being tested?

-Is the sample size sufficient to ensure adequate power to address the hypothesis being tested?

-Were correct statistical analysis used to support conclusions?

-Are there concerns about ethical or regulatory requirements being met?

Reviewer #1: The work is good and is presented well.

Reviewer #2: More details of the assumptions on data and the statistical testing procedure should be provided.

Reviewer #3: Objectives and Hypothesis: The study's objective—evaluating clinical agents as emergency treatments for B. multicinctus envenomation—is clearly stated. However, the hypothesis linking γ-bungarotoxin to severe clinical outcomes lacks explicit formulation.

Study Design: The preclinical mouse model appropriately tests toxin/drug interactions but omits critical design elements:

No randomization/blinding procedures described.

Dosage selection (e.g., 0.05 mg/kg atropine) lacks pharmacokinetic justification for human translation.

Sample Size and Power: Absence of sample size calculation or power analysis undermines statistical validity. For example, crude venom experiments showing "modest survival prolongation" may be underpowered.

Statistical Methods: Statistical tests (e.g., t-test vs. ANOVA) and software are unspecified. Reporting only P-values without effect sizes or confidence intervals limits interpretability.

Ethical Compliance: Institutional Animal Care and Use Committee (IACUC) approval permit number is missing.

Reviewer #4: -The objective: to determine whether early administration of clinically standard anticholinergic/anticholinesterase drugs could

protect patients or prolong survival during the pre-antivenom period, without interfering with subsequent antivenom efficacy.

- the study design appears appropriate to test the hypothesis, however there is a significant lack of detail in the methods section, making assessment and reproducibility difficult. Furthermore, it isn't clear if there were control groups and the only outcome measured was survival, no morbidity outcomes.

-there is no data analysis section in the methods and therefore no description of data analysis

**Results**

-Does the analysis presented match the analysis plan?

-Are the results clearly and completely presented?

-Are the figures (Tables, Images) of sufficient quality for clarity?

Reviewer #1: The results are clear

Reviewer #2: The results can be organized better so that the major findings are clear. Graphs can be presented concisely to better compare the treatments with respective toxin combinations.

Reviewer #3: Analysis Alignment: Results for purified toxins align with the methods, but crude venom experiments (showing non-significant effects) lack mechanistic follow-up to address this critical gap.

Data Presentation:

Figures/Tables referenced in-text (e.g., Table 1, Figure 2) are absent from the submitted draft.

Survival curves for toxin combinations are described but not visualized.

Statistical Reporting: Inconsistent reporting (e.g., P < 0.01 for toxins vs. P > 0.05 for crude venom) requires full statistical detail for verification.

Reviewer #4: - analysis plan wasn't described in the methods

- figures lack clear description and titles are too vague, difficult for the reader to understand what is what

-the results section has graphs, but doesn't contain the exact data and p-value (e.g. 61 mins vs. 50 mins p=0.03) in the text, just saying "significant protection"

-also there are a lot of figures, I wonder if it might be more fitting in a table? Or combined figures?

- interpretation of results should be in the discussion section

-is there a control group? If not, this is a limitation

**Conclusions**

-Are the conclusions supported by the data presented?

-Are the limitations of analysis clearly described?

-Do the authors discuss how these data can be helpful to advance our understanding of the topic under study?

-Is public health relevance addressed?

Reviewer #1: Conclusions are well supported by the data presented

Reviewer #2: Not needing Non-Parametric Statistical tests should be convinced to the reader before making conclusions.

γ-bungarotoxin was the primary focus of this study. However, the protective effects from Neostigmine methylsulfate

and atropine sulfate is minimally discussed in the manuscript. Given the extensive data set more analysis can be done on this respect.

Reviewer #3: Support from Data: Claims about clinical utility are overstated given the lack of efficacy against whole venom. The conclusion that drugs "provide supplementary benefits" is unsupported by crude venom data.

Limitations: Critical limitations are unaddressed:

Translational relevance of mouse-to-human dosing.

Inability of drugs to neutralize crude venom despite toxin efficacy.

Public Health Relevance: The focus on remote settings is well-placed, but failure to demonstrate whole-venom efficacy reduces practical impact

Reviewer #4: It is difficult to assess and interpret the results without the details of the study design and procedures.

Limitations are missing

**Editorial and Data Presentation Modifications?**

Reviewer #1: The work is good and is presented well. But need some minor revisions

1. Kindly spell check the entire document. Even snake was misspelled as 'snak' in keywords.

2. Most of the paragraphs starts with 'To...', which is usually not alowed in research papers.

Reviewer #2: (No Response)

Reviewer #3: Data Availability: Statement must specify repository (e.g., Dryad/Zenodo) and accession DOI per PLOS policy.

Figures/Tables: All referenced items must be included with clear legends.

Dosage Rationale: Add a table comparing mouse doses to human clinical ranges.

Mechanistic Clarification: Use subsections in Results to distinguish toxin-specific vs. venom-wide outcomes.

Reviewer #4: The manuscript is generally written in understandable English, but there are some minor issues such as typos, incorrectly and overused used colons and incorrect capitalization. Also, line numbers are missing from the manuscript, ,making it difficult for reviewers to give exact examples.

I would recommend the following:

Introduction

-adding some information about local burden of snake bites (morbidity and mortality), also are there DALYs for snake bites?

-giving background regarding current state of evidence for the use of these drugs for snake bites as well as current global and local recommendations/guidelines for the use of anticholinergic/anticholinesterase drugs in snake envenomation

Methods

The methods require more detail in order for readers to assess the methods, understand the results and for reproducibility. The details of the study design and procedures are not well-described.

- Manufacturer and location of headquarters should be mentioned for materials (venom and drugs)

- I would recommend making the subtitles about the studies more succinct, maybe something like: Pre-exposure (or maybe “prophylactic anticholinergic and anticholinesterase”?) anticholinergic and anticholinesterase drugs, Post- exposure anticholinergic and anticholinesterase drugs and Delayed post-exposure anticholinergic and anticholinesterase drugs

- Section “Protective effects study of pre-administered bungarotoxins needs work, sentences are not complete?

- Is it appropriate to only measure time of survival? What about other effects of venom like hemorrhage? This should be discussed in the discussion as a limitation

-More information needed regarding the procedures:

o how was time of survival measured? What outcomes?

o A data analysis section should be included.

o What were the exact procedures, e.g. groups studied? For example in the third group, were there two groups of 6 mice, one treated with neostigmine methylsufate and one with atropine sulfate? Where were the SC injections injected, the abdomen? How were the venoms/drugs stored? Which drugs IV, which SC and which IM, what was the process for IV admin for example. Why were some drugs IV and some IP? What were the procedures for each group?

**Summary and General Comments**

Reviewer #1: The work is good and is presented well. But need some minor revisions

1. Kindly spell check the entire document. Even snake was misspelled as 'snak' in keywords.

2. Most of the paragraphs starts with 'To...', which is usually not alowed in research papers.

Reviewer #2: The literature review can be elaborated more.

Results can be further discussed by related the results to Human Physiological conditions.

Reviewer #3: Strengths:

Clinically significant focus on accessible emergency treatments for resource-limited regions.

Rigorous evaluation of drug-toxin interactions with clear implications for adjunct therapy.

Weaknesses Requiring Major Revision:

Statistical Rigor: Specify tests, software, and effect sizes; justify sample sizes.

Translational Gap: Correlate mouse dosages to human safety/efficacy thresholds.

Crude Venom Efficacy: Address non-significance via:

Synergistic drug combinations.

Pre-treatment experimental designs.

Ethical Compliance: Provide IACUC permit number.

Novelty and Impact: The repurposing of FDA-approved drugs is innovative and aligns with PLOS NTDs' scope for practical solutions in tropical medicine. With revisions, this work could significantly impact snakebite management protocols.

Reviewer #4: This is an interesting investigation into the effects of anticholinergic/antocholinesterase drugs pre- and post- snake envenomation. As an official NTD, it is helpful to have more information on management of morbidity and mortality due to snake bites.

The manuscript requires work to be suitable for publication, especially in describing the study design and procedures in the methods. Furthermore, the study does need to be placed in context in the introduction, as it seems there have already been studies on the use of anticholinergic/antocholinesterase drugs in snake envenomation and there are guidelines about there use for this purpose (https://pmc.ncbi.nlm.nih.gov/articles/PMC11402871/).

PLOS authors have the option to publish the peer review history of their article (what does this mean? ). If published, this will include your full peer review and any attached files.

**Do you want your identity to be public for this peer review?** For information about this choice, including consent withdrawal, please see our Privacy Policy .

Reviewer #1: **Yes: ** Gopal Samy B

Reviewer #2: No

Reviewer #3: No

Reviewer #4: No

**Figure resubmission:**

**Reproducibility:**



---

## [Decision Letter · Decision Letter 1]

15 Oct 2025

Protective Effects of Clinical Anticholinergic and Anticholinesterase Agents Against Bungarus multicinctus Venom and Neurotoxin-Rich Snake Venoms

Dear Dr. Lee,

Thank you for submitting your manuscript to PLOS Neglected Tropical Diseases. After careful consideration, we feel that it has merit but does not fully meet PLOS Neglected Tropical Diseases's publication criteria as it currently stands. Therefore, we invite you to submit a revised version of the manuscript that addresses the points raised during the review process.

Please submit your revised manuscript within 60 days Nov 14 2025 11:59PM. If you will need more time than this to complete your revisions, please reply to this message or contact the journal office at plosntds@plos.org. Please include the following items when submitting your revised manuscript:

We look forward to receiving your revised manuscript.

Kind regards,

Wuelton Monteiro, Ph.D.

Section Editor

Wuelton Monteiro

Section Editor

Shaden Kamhawi

co-Editor-in-Chief

Paul Brindley

co-Editor-in-Chief

**Journal Requirements:**

At this stage, the following Authors/Authors require contributions: Guowen Zhang, Mengqi He, Tongyi Sun, and Wen-Hui Lee. Please ensure that the full contributions of each author are acknowledged in the "Add/Edit/Remove Authors" section of our submission form.

**Reviewers' Comments:**

Reviewer's Responses to Questions

**Key Review Criteria Required for Acceptance?**

**Methods**

-Are the objectives of the study clearly articulated with a clear testable hypothesis stated?

-Is the study design appropriate to address the stated objectives?

-Is the population clearly described and appropriate for the hypothesis being tested?

-Is the sample size sufficient to ensure adequate power to address the hypothesis being tested?

-Were correct statistical analysis used to support conclusions?

-Are there concerns about ethical or regulatory requirements being met?

Reviewer #2: (No Response)

Reviewer #3: The study employs well-defined animal models with clear descriptions of drug and venom administration to evaluate protective effects. Randomization by body weight is a strength that reduces bias. Statistical analyses are appropriate, and ethical standards are met. However, the use of high venom doses (2× or 3× LD₅₀) may produce overly severe effects that do not fully replicate real snakebite conditions. Additionally, the rationale for the selected time intervals between drug and venom injections is not adequately justified, limiting the clinical relevance of the findings. Overall, while the methodology is sound, providing justification for dose selection and timing would enhance the study’s translational applicability.

**Results**

-Does the analysis presented match the analysis plan?

-Are the results clearly and completely presented?

-Are the figures (Tables, Images) of sufficient quality for clarity?

Reviewer #2: (No Response)

Reviewer #3: The results follow the planned analysis and clearly show the effects of the drugs tested. The use of survival times is straightforward, but adding other measures could strengthen the conclusions. The figures are clear, but referring too much to supplementary data makes it a bit hard to follow. Overall, the results are well presented but could be easier to understand with fewer cross-references.

**Conclusions**

-Are the conclusions supported by the data presented?

-Are the limitations of analysis clearly described?

-Do the authors discuss how these data can be helpful to advance our understanding of the topic under study?

-Is public health relevance addressed?

Reviewer #2: (No Response)

Reviewer #3: The conclusions are well supported by the data, showing certain drugs protect against B. multicinctus venom. The authors clearly mention limitations, such as the drugs’ negative effects on other snake venoms. They explain how these findings improve understanding of snakebite treatment, especially when antivenom is not immediately available. The study also highlights important public health issues for rural areas lacking quick access to antivenom.

**Editorial and Data Presentation Modifications?**

Reviewer #2: (No Response)

Reviewer #3: The Methods and Results sections are generally clear and well-structured, providing detailed descriptions and relevant data. To enhance clarity, it’s recommended to briefly justify key experimental choices such as venom doses and timing of drug administration in Methods. In Results, ensure all figures and tables are clearly labeled with consistent terminology and units. Adding brief explanations of statistical tests used would strengthen transparency. Simplifying complex sentences and avoiding jargon will improve readability. Overall, minor editorial refinements here will make the sections more accessible and scientifically robust.

**Summary and General Comments**

Reviewer #2: (No Response)

Reviewer #3: (No Response)

PLOS authors have the option to publish the peer review history of their article (what does this mean? ). If published, this will include your full peer review and any attached files.

**Do you want your identity to be public for this peer review?** For information about this choice, including consent withdrawal, please see our Privacy Policy .

Reviewer #2: No

Reviewer #3: **Yes: ** Dr Jigna Gohil

**Figure resubmission:**

**Reproducibility:**



---

## [Editor Report · Decision Letter 2]

17 Nov 2025

Dear Ms. Lee,

We are pleased to inform you that your manuscript 'Protective Effects of Clinical Anticholinergic and Anticholinesterase Agents Against Bungarus multicinctus Venom and Neurotoxin-Rich Snake Venoms' has been provisionally accepted for publication in PLOS Neglected Tropical Diseases.

Best regards,

Wuelton Monteiro, Ph.D.

Section Editor

Wuelton Monteiro

Section Editor

Shaden Kamhawi

co-Editor-in-Chief

Paul Brindley

co-Editor-in-Chief

---

## [Editor Report · Acceptance letter]

Dear Ms. Lee,

We are delighted to inform you that your manuscript, " 

Protective Effects of Clinical Anticholinergic and Anticholinesterase Agents Against Bungarus multicinctus Venom and Neurotoxin-Rich Snake Venoms," has been formally accepted for publication in PLOS Neglected Tropical Diseases.

Best regards,

Shaden Kamhawi

co-Editor-in-Chief

Paul Brindley

co-Editor-in-Chief
